# OpenReview forum: "Overcoming Vocabulary Constraints with Pixel-level Fallback"
_ICML.cc/2025/Workshop/TokShop — TokShop_

### Official Review · Reviewer_Un8G · 2025-06-06
**Reasonable text encoding approach that could still benefit from further investigation**

**Rating:** 7
**Confidence:** 4

**Review:**

The authors propose a pixel-based text (represented as images) encoding method as a way to tackle out-of-vocabulary or under-resourced vocabulary related problems often present in multilingual scenarios that include varying writing scripts. The proposed method should improve the input text compression, leading to faster inference times as demonstrated in the experiments section while maintaining at least the same performance quality (e.g. in MT) compared to the LLMs with the vanilla text tokenizer/encoder. They also demonstrate that the pixel-based method complements the models with closed-based vocabulary such as BERT.

I have several questions/suggestions regarding the experiments.

I understand the choice of the models, ranging from small SmolLL2 to slightly larger Phi-3-mini, however, I think that the experiments require better choice of baselines to determine whether the improvements brought by the +Pixel models are due to the embedding method itself or due to increasing the overall number of model parameters. If I understood correctly, the Pixel encoder consists of roughly 100M parameters, expanding the SmolLL2-360M by 1/4 of the original model size inviting the question whether the SmolLL2 performance would not also increase by a similar amount by simply increasing the number of parameters of said base model. With larger models, the performance gains of the +Pixel versions are smaller possibly due to the Pixel encoder providing much smaller proportional increase. This could be possibly refused by Table 1, however, it is not clear why the authors decided to fine-tune the models in Table 1 for only a single epoch and not rather aiming at early-stopping. Is it possible to provide a clarification?

Similarly, have you tried tuning the models for longer than 1000 steps in the experiments summarized in Table 2? It is not clear from the table whether the model performance plateaued at the 1000-step mark or whether the models were still improving.

Table 5 could benefit from a more detailed ablation study. How did you choose the fallback network hyper-parameter combinations in this table?

There seems to be a mismatch between the values in Table 7 and the text saying "[...] Notably, pretraining and finetuning with prefix tokens (prefix) reduces the distancebetween centers (2.6 vs. 40.7) [...]" (the value 40.7 is missing from the table.

---

### Official Review · Reviewer_6fc7 · 2025-06-08

**Rating:** 8
**Confidence:** 5

**Review:**

This paper introduces a pixel-based fallback network to address vocabulary limitations in English-centric language models when processing non-Latin scripts. The core innovation is augmenting pretrained language models with a vocabulary-free encoder that generates embeddings from text rendered as images, bypassing tokenizer constraints for underrepresented languages and scripts.

Pros:
- The paper is well written and easy to follow.
- The approach is technically sound, building on established Vision Transformer architectures adapted for text rendering. The fallback network design is sensible, using character bigrams as patches and employing attention masking to process words independently while enabling batch processing for efficiency.
- Results demonstrate consistent improvements across settings, with particularly impressive gains for smaller models. The finding that a pixel-augmented 360M model can outperform unmodified larger models (1.7B on Thai translation) is noteworthy and practically valuable.


Cons:

-  Experiments focus primarily on translation tasks with English as the target language. The approach would benefit from evaluation on more diverse tasks and language pairs, particularly those not involving English.

- The paper would benefit from deeper analysis of when and why the pixel-based approach underperforms, particularly for the Phi-3-mini results in topic classification.

---

### Decision · Program_Chairs · 2025-06-10

Accept